# An environmental justice analysis of air pollution emissions in the United States from 1970 to 2010

Yanelli Nunez [1,2] ✉, Jaime Benavides[1], Jenni A. Shearston[1,3], Elena M. Krieger[2], Misbath Daouda[1,3], Lucas R. F. Henneman [4], Erin E. McDuffie[5], Jeff Goldsmith[6], Joan A. Casey[1,7] & Marianthi-Anna Kioumourtzoglou[1]

Over the last decades, air pollution emissions have decreased substantially; however, inequities in air pollution persist. We evaluate county-level racial/ethnic and socioeconomic disparities in emissions changes from six air pollution source sectors (industry [$SO_2$], energy [$SO_2$, $NO_x$], agriculture [$NH_3$], commercial [$NO_x$], residential [particulate organic carbon], and on-road transportation [$NO_x$]) in the contiguous United States during the 40 years following the Clean Air Act (CAA) enactment (1970-2010). We calculate relative emission changes and examine the differential changes given county demographics using hierarchical nested models. The results show racial/ethnic disparities, particularly in the industry and energy generation source sectors. We also find that median family income is a driver of variation in relative emissions changes in all sectors—counties with median family income >\$75 K vs. less generally experience larger relative declines in industry, energy, transportation, residential, and commercial-related emissions. Emissions from most air pollution source sectors have, on a national level, decreased following the United States CAA. In this work, we show that the relative reductions in emissions varied across racial/ethnic and socioeconomic groups.

The United States (US) has seen reductions in air pollution emissions from various source sectors since the enactment of the Clean Air Act (CAA) in 1970, contributing to improving air quality substantially[1–3]. However, studies show that racial/ethnic and socioeconomic inequities in air pollution exposure persist across the US despite the nationwide downward trend in air pollution[4–7], indicating inequities in air pollution emissions reductions.

Environmental justice studies in air pollution have heavily relied on modeled ambient air pollution concentrations[4–6,8–10]. Unlike measured air pollutant concentrations, modeled concentrations can more comprehensively cover geographical regions, including those where air quality monitors are sparse[11]. However, pollution concentrations alone do not provide information about specific air pollution sources contributing to the observed disparities. A couple of studies have addressed this knowledge gap by modeling source-specific air pollution concentrations[7,12]. The disconnect between the studied air pollution concentrations and air pollution sources presents a barrier to developing efficient and economically feasible regulatory strategies to address air pollution inequities. Equitable emissions decrease, i.e., greater reductions for overburdened

[1]Dept. of Environmental Health Sciences, Columbia University Mailman School of Public Health, New York City, NY, USA. [2]PSE Healthy Energy, Oakland, CA, USA. [3]Dept. of Environmental Science, Policy, & Management, University of California Berkeley School of Public Health, Berkeley, CA, USA. [4]Sid and Reva Dewberry Dept. of Civil, Environmental, and Infrastructure Engineering, George Mason University, Fairfax, VA, USA. [5]Dept. of Energy, Environmental & Chemical Engineering, Washington University in St. Louis, St Louis, MO, USA. [6]Dept. of Biostatistics, Columbia University Mailman School of Public Health, New York City, NY, USA. [7]Dept. of Environmental and Occupational Health Sciences, University of Washington, Seattle, WA, USA. ✉e-mail: y.nunez@psehealthyenergy.org

groups, can then facilitate a more just reduction of air pollution concentrations and exposures.

Most evidence on targetable air pollution sources comes from cross-sectional analyses. Several studies have used residential proximity to pollution sources as a metric to evaluate inequities. These studies have found racial/ethnic and socioeconomic disparities in the spatial distribution of industrial facilities[6,13], landfills, hazardous waste sites[14,15], gas[16] and coal-fired power plants[17], roadways[18], and other pollution sources[19,20]. Residential proximity studies are generally cross-sectional and often focus on a single air pollution source sector. Other studies have evaluated inequity in multiple air pollution sources by leveraging data from local emissions inventories, but these analyses also focused on a single time point[7]. Thus, they do not provide information about temporal inequity trends in emissions changes.

In this work, we evaluated county-level racial/ethnic and socioeconomic disparities in air pollution emissions changes in the contiguous US from 1970 to 2010 in the transportation (nitrogen oxides [$NO_x$]), agriculture (ammonia [$NH_3$]), residential (particulate organic carbon), commercial ($NO_x$), industry (sulfide dioxide [$SO_2$]), and energy ($NO_x$ and $SO_2$) sectors. We used county-level data on race/ethnicity and socioeconomic status (SES) from the decennial Census (1970–2010)–race/ethnicity group definitions were based on the Census definition in each decennial. We leveraged air pollution emissions data from the Community Emissions Data Global Burden of Disease Map ($CED_{GBD-MAP}$) to estimate county-level relative emissions changes for the six air pollution source sectors, using specific pollutant tracers per source sector. $CED_{GBD-MAP}$ is an air pollution emissions inventory that uses emissions data from local and regional inventories and activity data to calculate gridded emissions estimates for the globe from 1970 to 2017[21]. Using hierarchical models, we evaluated disparities in emissions changes by modeling the association between county-level demographics and the decennial relative change in emissions. We focused on relative emissions changes because, to support equity, emission decreases should be relative to existing emissions levels in each county[20]. That is, places with higher pollution levels should have higher emissions reductions. We found racial/ethnic disparities in emissions changes, particularly in the industry and energy sector. We also found that median family income is a driver of emissions changes in all air

pollution source sectors–counties with median family income >$75 K experienced larger declines in industry, energy, transportation, residential, and commercial-related emissions.

## Results

### Trends in demographics
From 1970 to 2000, the US grew in racial/ethnic diversity. Nationwide county average Hispanic population percentage increased from 3.2 to 6.2%, the Asian population from 0.3 to 3.4%, and the American Indian population from 0.9 to 1.6%, whereas the percentage Black population decreased from 9.1 to 8.8%. During the study time, the average county White population percentage was 87.5%. The average county percent poverty decreased from 20.4 to 13.7%, but unemployment increased from 4.5 to 5.8%. Average county-level median family income (2010 adjusted dollars) increased from $45,400 to $55,100 and the median property value (2010 adjusted dollars) increased from $33,700 to $110,200. Table 1 presents summary statistics for the demographic variables at each decennial included in the study (1970, 1980, 1990, and 2000) and Supplementary Fig. 1 shows the Spearman correlation coefficients among these variables.

### Trends in air pollution emissions
We observed variability in emission changes across counties with variation in both the magnitude and direction of change (Fig. 1 and Supplementary Dashboard). On average, air pollution emissions across the US decreased substantially from 1970 to 2010 from all source sectors we considered except for agriculture $NH_3$ and residential particulate organic carbon (OC). We observed the most pronounced emission decreases for $SO_2$ from industry and energy, which fell from a mean of 5.6 to 0.6 (−89.3%) and 9.0 to 3.0 (−66.7%) kg/km²/day, respectively, over the 40 years. Nationwide average emissions of $NO_x$ from transportation and energy decreased more moderately from 5.2 to 2.2 (−57.7%) and 2.5 to 1.5 (−40.0%) kg/km²/day, respectively. $NO_x$ emissions from commercial sources decreased from 0.3 to 0.2 (−50.0%) kg/km²/day. Nationwide average emissions of $NH_3$ from agriculture increased from 0.7 to 1.3 (85.7%) kg/km²/day, and OC emissions from the residential source sector remained constant at 0.1 kg/km²/day (Table 2 and Supplementary Fig. 2). The average decennial relative emission changes for each of the evaluated air pollutants are presented in Table 3.

## Table 1 | Variables of interest and potential confounders

|  | 1970, N = 3105 | 1980, N = 3108 | 1990, N = 3110 | 2000, N = 3109 |
|---|---|---|---|---|
| **Variables of interest** |  |  |  |  |
| % White | 89.7 (86.7, 99.4) | 88.4 (83.1, 98.8) | 87.4 (81.0, 98.4) | 84.8 (77.2, 96.7) |
| % Black | 9.1 (0.1, 11.3) | 8.7 (0.1, 10.4) | 8.7 (0.2, 10.1) | 8.8 (0.3, 10.1) |
| % Asian | 0.3 (0.1, 0.3) | 0.4 (0.1, 0.4) | 2.4 (0.3, 1.9) | 3.4 (0.6, 3.5) |
| % American Indian | 0.9 (0.0, 0.2) | 1.2 (0.1, 0.5) | 1.5 (0.2, 0.6) | 1.6 (0.2, 0.8) |
| % Hispanic | 3.2 (0.0, 1.9) | 3.8 (0.5, 1.8) | 4.5 (0.4, 2.5) | 6.2 (0.9, 5.1) |
| % Poverty | 20.4 (12.1, 26.3) | 15.4 (10.3, 19.0) | 16.2 (10.9, 19.9) | 13.7 (9.2, 16.8) |
| % Unemployment | 4.5 (3.0, 5.5) | 6.8 (4.5, 8.5) | 6.6 (4.6, 8.1) | 5.8 (4.0, 6.9) |
| Median Family Income (×$1000) | 45.4 (38.2, 51.2) | 46.9 (40.1, 52.9) | 48.2 (40.5, 53.8) | 55.1 (46.8, 60.9) |
| Median Property Value (×$1000) | 33.7 (23.2, 40.8) | 98.3 (72.9, 113.0) | 91.3 (62.5, 101.8) | 110.2 (75.7, 126.3) |
| **Covariates** |  |  |  |  |
| Population density (population/km²) | 82.5 (5.6, 28.2) | 83.7 (6.3, 33.8) | 86.0 (6.2, 35.7) | 93.3 (6.7, 39.8) |
| **Urbanicity** |  |  |  |  |
| Metropolitan | 675.0 (21.7%) | 788.0 (25.4%) | 836.0 (26.9%) | 907.0 (29.2%) |
| Micropolitan | 1577.0 (50.8%) | 1583.0 (50.9%) | 1537.0 (49.4%) | 1525.0 (49.1%) |
| Non-urban | 853.0 (27.5%) | 737.0 (23.7%) | 737.0 (23.7%) | 677.0 (21.8%) |

County-level annual mean (interquartile range, IQR) for continuous variables and percent observations for the categorical variables. N = total number of counties at each decennial. Race/ethnicity percentages do not add up to 100% because race groups include Hispanic and non-Hispanic populations.

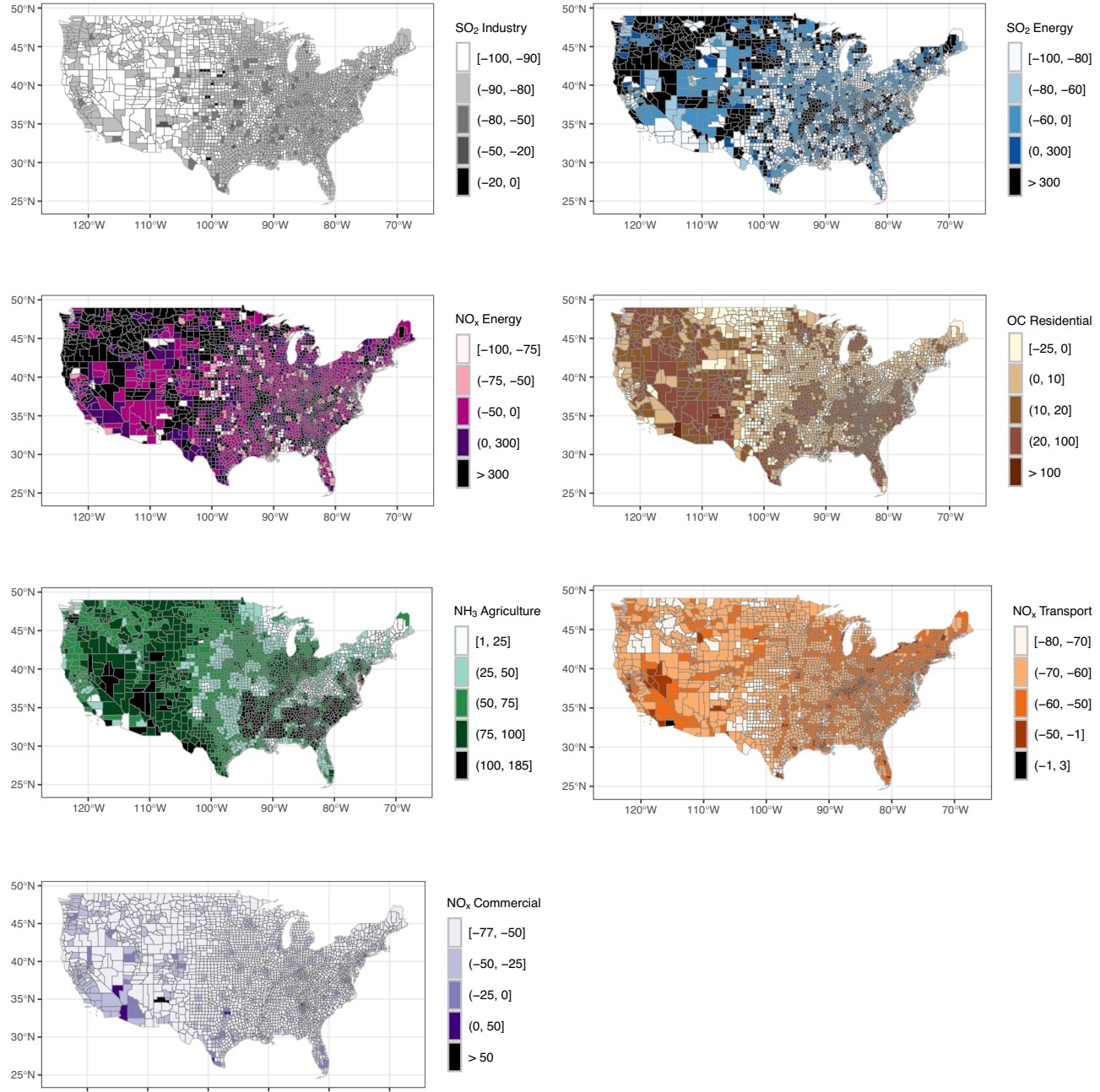

**Fig. 1 | Pollutant emissions: County-level relative decennial emissions changes (percentages) from 1970 to 2010 for the seven air pollutants evaluated in the study.** Darker tone colors indicate larger relative emissions increases or lower relative emissions decreases. See Supplementary Dashboard for maps of emissions (kg/km²/day) for each pollutant and during each year. This figure used county spatial shape files obtained from NHGIS.

## County-level disparities by racial/ethnic groups and socioeconomic status

In a longitudinal analysis from 1970 to 2010, we evaluate the relationship between relative decennial emissions changes and racial/ethnic groups (White, Black, Asian, American Indian, Hispanic) as well as socioeconomic indicators (median family income, median property value, percent poverty, percent unemployment) at the county level across the United States. Figures 2 (racial/ethnic) and 3 (socioeconomic) show all the association curves and we describe our results below. Numeric effect estimates for the linear associations are in Supplementary Table 1. Note that for both linear and nonlinear associations, a positive association indicates percentage point increases in the relative emissions change—i.e., larger relative emissions increases

(e.g., from 5 to 10%) or smaller emissions decreases (e.g., from −10 to −5%)—per increase in the demographic variable of interest. Conversely, a negative association indicates percentage point decreases in the relative emission change—i.e., smaller increases (e.g., from 10 to 5%) or more prominent decreases (e.g., from −5 to −10%)—per increase in the demographic variable of interest.

### Industry: SO₂ emissions

The association between each racial/ethnic group and the relative decennial change in industrial SO₂ emissions was nonlinear, except for the Black population percentage. We found a −1.35 percentage point (pp) (95% CI: −2.09, −0.60) decrease in the industrial SO₂ relative emissions change per a 10 pp increase in the proportion of county-

**Table 2 | Air pollution emissions**

| Air pollution source sector (kg/km²/day) | 1970, N = 3101 | 1980, N = 3103 | 1990, N = 3105 | 2000, N = 3104 | 2010, N = 3103 |
|---|---|---|---|---|---|
| Industry: $SO_2$ | 5.6 (0.0, 3.2) | 3.2 (0.0, 1.7) | 2.6 (0.0, 1.3) | 1.4 (0.0, 0.8) | 0.6 (0.0, 0.4) |
| Energy: $SO_2$ | 9.0 (0.0, 5.9) | 9.1 (0.0, 5.9) | 8.5 (0.0, 5.0) | 6.2 (0.0, 3.8) | 3.0 (0.0, 1.6) |
| Energy: $NO_x$ | 2.5 (0.0, 2.0) | 3.5 (0.0, 2.5) | 3.5 (0.0, 2.6) | 2.8 (0.0, 2.3) | 1.5 (0.0, 1.3) |
| Agriculture: $NH_3$ | 0.7 (0.3, 1.0) | 0.9 (0.4, 1.2) | 1.0 (0.4, 1.2) | 1.1 (0.4, 1.5) | 1.3 (0.5, 1.7) |
| Transportation: $NO_x$ | 5.2 (3.4, 5.8) | 4.8 (3.0, 5.4) | 4.3 (2.7, 4.9) | 3.9 (2.2, 4.5) | 2.2 (1.2, 2.5) |
| Residential: OC | 0.1 (0.0, 0.1) | 0.2 (0.0, 0.1) | 0.1 (0.0, 0.1) | 0.1 (0.0, 0.1) | 0.1 (0.0, 0.1) |
| Commercial: $NO_x$ | 0.3 (0.0, 0.2) | 0.2 (0.0, 0.1) | 0.4 (0.0, 0.2) | 0.3 (0.0, 0.1) | 0.2 (0.0, 0.1) |

County-level annual mean (interquartile range, IQR) emissions. *N* = total number of counties included at the analyses for each time point.

**Table 3 | Air pollution relative emissions change**

| | Mean relative change | IQR |
|---|---|---|
| Transportation: $NO_x$ | −19.2 | −32.3, −8.9 |
| Agriculture: $NH_3$ | 13.8 | 8.8, 18.3 |
| Residential: OC | 9.8 | −31.6, 37.3 |
| Commercial: $NO_x$ | −14.3 | −33.6, 29.3 |
| Energy: $NO_x$ | 41.2 | −44.3, 41.1 |
| Energy: $SO_2$ | 68.1 | −49.4, 7.0 |
| Industry: $SO_2$ | −38.1 | −51.9, −19.3 |

County-level mean and interquartile range (IRQ) for the relative change (%) in emissions (2000–2010).

level Black population; we also found an overall negative nonlinear association between percent White population and the relative $SO_2$ emissions change. However, the proportions of American Indian and Hispanic populations were positively associated with the relative change in industrial $SO_2$ emissions in population percentages above ~35%. We found no clear association between county-level Asian population percentage and the change in industrial $SO_2$ emissions.

The associations between socioeconomic variables and the change in industrial $SO_2$ emissions were all nonlinear. Percent unemployment and poverty were positively associated with the relative decennial change in industrial $SO_2$ emissions, with stronger associations at higher percentages. In the case of median family income, the association was negative, with a stronger association below ~$50 K. We found no clear association between median property value and the relative change in industrial $SO_2$ emissions.

### Energy: $SO_2$ and $NO_x$ emissions
We found no associations between the race/ethnicity groups and the change in energy $SO_2$ emissions.

The association between each of the racial/ethnic groups and the change in energy $NO_x$ emissions were all linear except for the Hispanic population percentage. A 10 pp increase in the White or Black population percentage was associated with a −9.25 pp (95% CI: −15.01, −3.48) and −17.43 pp (95% CI: −23.88, −10.98) decrease in energy $NO_x$ relative emissions change, respectively. However, a 10pp increase in the Asian population percentage was associated with an 18.52 pp (95% CI: 7.41, 29.63) increase in energy $NO_x$ relative emissions change, and a 10 pp increase in the American Indian population percentage with an 11.78 pp (95% CI: 4.20, 19.38) increase. The Hispanic population was also positively associated with the change in energy $NO_x$ relative emissions change, with a slightly stronger association at high Hispanic population percentages.

The associations between socioeconomic variables and the change in energy $SO_2$ emissions were all nonlinear. Percent unemployment was positively associated with the relative decennial change in energy emissions below ~5% and null at higher unemployment percentages. Percent poverty was positively associated with the outcome,

and this association became stronger with increasing percent poverty levels. Median family income was negatively associated with the relative change in energy $SO_2$ emissions but the association plateaued above ~$50 K. The median property value was also negatively associated with the relative change in energy $SO_2$ emissions but only below ~$100 K; above that, there was no association.

The associations between socioeconomic variables and the change in energy $NO_x$ emissions were also all nonlinear. Percent poverty was positively associated with the relative change in energy $NO_x$ emissions below 20% poverty and negative above this value. Median family income was negatively associated with the relative change in $NO_x$ emissions above ~$50 K but null for lower income values. We found no association between percent unemployment or median property value with the relative emissions change in energy $NO_x$.

### Agriculture: $NH_3$ emissions
We found no associations between county-level White or Asian percentage population and the relative decennial change in agricultural $NH_3$ emissions. However, a 10 pp increase in the American Indian percentage population was linearly associated with a 0.33 pp (95% CI: −0.02, 0.67) increase in the relative change of agriculture $NH_3$ emissions. We also found a positive nonlinear association between the Hispanic population percentage and the relative change in agricultural $NH_3$ emissions, with steeper slopes in high county-level Hispanic population percentages. The Black population percentage was nonlinearly positively associated with the relative change in agricultural $NH_3$ emissions in population percentages below 50%. Above that percentage, the association was null.

The associations between socioeconomic variables and the change in agricultural $NH_3$ emissions were all nonlinear. We found a positive association between percent unemployment and the relative change in agricultural $NH_3$ emissions in unemployment levels below ~7% but a negative association at higher unemployment percentages. We found a negative association between percent poverty and the relative decennial change in $NH_3$ emissions. Conversely, median family income was positively associated with the relative change in $NH_3$ emissions, but the association plateaued and was null above $50 K. We also found a positive association between median property value and the relative decennial change in agriculture $NH_3$ emissions above ~$750 K.

### On-road transportation: $NO_x$ emissions
We found a negative nonlinear association between county-level Asian population percentage and the relative decennial change in transportation $NO_x$ emissions, but after ~25%, the association plateaued. We found no associations between the White, Black, American Indian, or Hispanic percentage population and the relative change in transportation $NO_x$ emissions.

The associations between each socioeconomic variable and the relative change in transportation $NO_x$ emissions were all nonlinear. We found that percent unemployment and population in poverty were

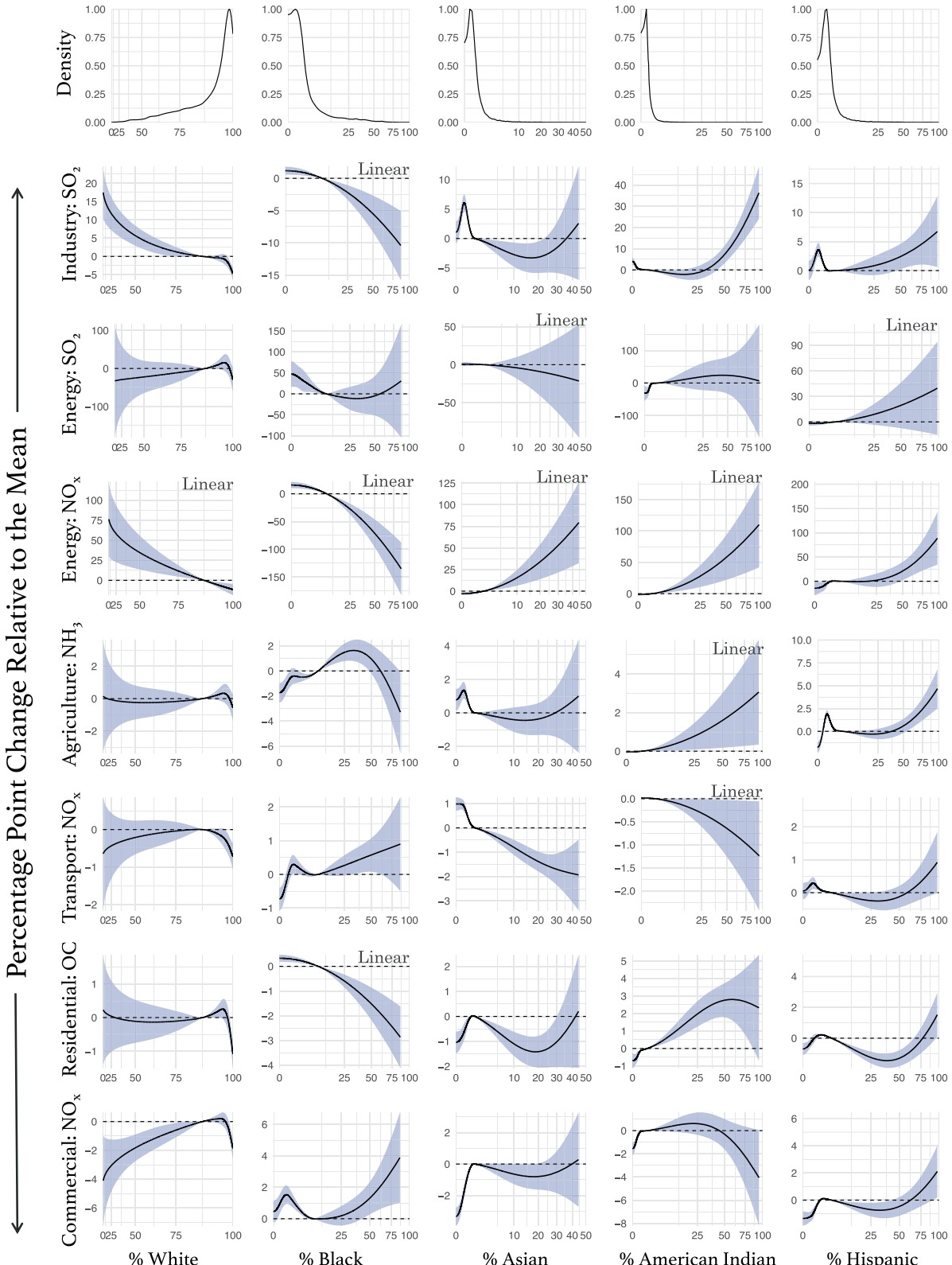

**Fig. 2 | Association curves for the racial/ethnic groups.** All models are adjusted for population density, urbanicity, EPA geographic region, and year. The x-axis was scaled to improve the visibility of the curve in the low/high x values (squared root for the percent White population and squared for the other demographics). The curves represent the relationship between the demographic variables and change in the relative emissions (percentage point difference) relative to the mean of the demographic variable (e.g., from the national Hispanic population average of 4.5% [during the study period] to 75%, energy NO$_x$ relative emissions increase 50 percentage points). The shaded area on the curve is the 95% confidence interval. Linear associations are indicated by the "Linear" label. Density plots for the independent variables are shown at the top of each column.

negatively associated with the relative change in transportation $NO_x$ emissions. For unemployment, the association was null above ~20% unemployment. The association between median family income and the relative decennial change in transportation $NO_x$ emissions was positive below ~$75 K and negative above this value. Median property value had a positive association with the relative change in transportation $NO_x$ emissions below ~$100 K.

### Residential: OC emissions
We found that a 10 pp increase in the county-level Black population percentage was linearly associated with a −0.37 pp (95% CI: −0.54, −0.19) decrease in the relative decennial change in residential OC emissions. White, Asian, American Indian, and Hispanic population percentages were nonlinearly associated with the relative change in residential OC emissions. The percentage of White population was negatively associated with the relative change in residential OC emissions in percentages above ~85% White population. Below that, the association was null. In the Asian and Hispanic population percentages, the association was positive below ~5% and mostly negative above. The American Indian population percentage was positively associated with the relative change in residential OC emissions but the association plateaued above 50%.

The associations between socioeconomic variables and the relative decennial change in residential OC emissions were all nonlinear. Percent unemployment was positively associated with the relative change in residential OC emissions below 5% and null in higher percentages of unemployment. The association is positive below ~10% and negative at higher values for the percent below poverty. Median family income was overall negatively associated with the relative change in residential OC emissions but the association was more robust at values above $75 K. The median property value was positively associated with the relative change in OC emissions below ~$125 K and negative at higher median property values.

### Commercial: $NO_x$ emissions
The associations between racial/ethnic groups and the relative decennial change in commercial $NO_x$ emissions were nonlinear. We found a positive association in county-level White population percentages below 80% and a negative association above that. In the case of the Black population percentage, the association was overall positive. For the Asian percentage population, the association was positive below ~5% and null above that percentage. We found no association with the American Indian percentage population. The Hispanic population percentage was positively associated with the relative change in commercial $NO_x$ emissions in population percentages below ~5% and null after that.

The associations between socioeconomic variables and the change in $NO_x$ emissions were all nonlinear. Unemployment was positively associated with the relative decennial change in commercial $NO_x$ emissions below 5% and negative at higher values. Percent poverty and median property value were overall negatively associated with the relative change in commercial $NO_x$ emissions. In the case of percent poverty, the association plateaued after ~20% and for property value, the association was stronger after ~$500 K. The association between median family income and the relative change in commercial $NO_x$ emissions was positive below ~$75 K and negative above that.

We have summarized all results described above in Fig. 4.

### Confounding by socioeconomic status
We also evaluated the robustness of our results for the racial/ethnic groups to potential confounding bias by SES. In this sensitivity analysis, we modeled the associations for each racial/ethnic group, adjusting for socioeconomic variables (median family income, property value, percent poverty, and percent unemployment). The results from the sensitivity analysis, for the most part, support the primary analysis results,

but we did observe some differences, particularly in the residential and commercial sectors. In the sensitivity analysis, the association between the percent White population and the change in residential OC and commercial $NO_x$ emissions was null. In contrast, these associations were negative in the main analysis. We also did not find an association between the Black and Asian population percentages with the change in emissions from these two air pollution source sectors. Furthermore, we did not find positive associations in the lower population percentages for the Hispanic population as in the main analysis. The results from this sensitivity analysis are presented in Supplementary Fig. 3.

### Regional analyses
To evaluate the relevance of national results to sub-national areas, we constructed regional models for industry $SO_2$ and transportation $NO_x$ by EPA regions. The regional models evaluated the association between median family income, percent Black population, and percent White population with the relative decennial change in emissions in each of these two air pollution source sectors. For the most part, the regional results agreed with the national-level results, but there were some differences, particularly for the racial/ethnic groups. In the national analysis, we found no association between the White and Black population percentages and the relative change in transportation $NO_x$ emissions; however, in regions 1–3 (Northeast area), 5 (portions of upper Midwest and Ohio Valley), and 6 (southern Texas), we found a clear positive association between the Black population percentage and the relative change in transportation $NO_x$ emissions (Supplementary Fig. 4). We also found a positive association between the White population percentage and the relative change in transportation $NO_x$ emissions in Region 1–3 (Northeast area). In the case of $SO_2$ emissions from industry, for the White population percentage, we found a negative association in Region 8 (Northern Rockies and Plains area), which reflected the results from the main analysis; however, in regions 1–3 (Northeast area) the association was positive and null in the rest of the regions. For the Black population percentage, we found a positive association with the relative change in industry $SO_2$ emissions in Regions 1–3—opposite to the national analysis results, which showed a negative association—and no association in other EPA regions (Supplementary Fig. 5). It is important to note that splitting the data into regions decreased statistical power, likely influencing our ability to detect associations in some of the regions.

## Discussion
Our research provides a national investigation of air pollution emission changes in the 40 years following the CAA enactment. We examined racial/ethnic and socioeconomic disparities in the relative decennial emissions change for six air pollution source sectors: industry, energy, agriculture, on-road transportation, commercial, and residential. In our study, we found that socio-demographic characteristics of counties were associated with the county-level relative changes in air pollution emissions from 1970 to 2010. Our results suggest many, albeit not universally, racial/ethnic and socioeconomic inequalities in reducing air pollution emissions following the CAA.

Emissions from energy generation had some of the largest emissions reductions among the air pollution source sectors we analyzed. However, we found that energy $SO_2$ relative emissions changes had prominent socioeconomic disparities. We also found racial/ethnic inequities in energy $NO_x$ relative emissions changes. The 1990 CAA amendment introduced the Acid Rain Program (ARP), setting a cap on total $SO_2$ levels to 50% of 1980 levels by 2010 and a two million ton reduction in $NO_x$ emissions by 2000 (no cap for $NO_x$)[22]. The ARP introduced the allowance trading system that uses market-based incentives to reduce air pollution. For each ton reduced below the applicable emissions limit, owners of a generating unit received an emissions allowance they could use at another unit, keep for future use, or sell[22]. CAA market-based policy strategies have helped achieve

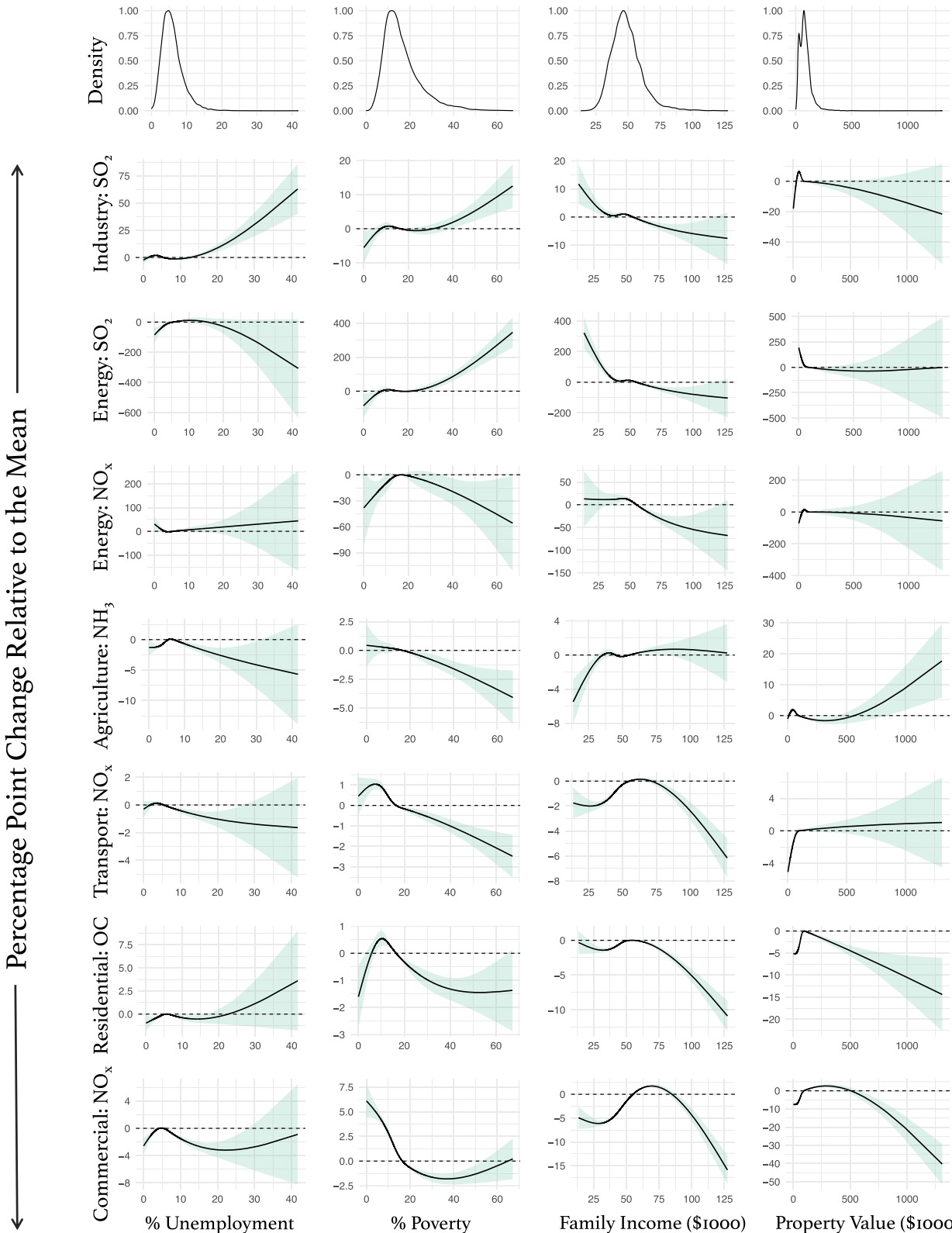

**Fig. 3 | Association curves for the socioeconomic variables.** All models are adjusted for population density, urbanicity, EPA geographic region, and year. The curves represent the relationship between the demographic variables and change in the relative emissions (percentage point difference) relative to the mean of the demographic variable (e.g., from the national poverty average of 16.5% [during the study period] to 40%, energy $SO_2$ relative emissions increase 100 percentage points). The shaded area on the curve is the 95% confidence interval. All associations were nonlinear. Density plots for the independent variables are shown at the top of each column.

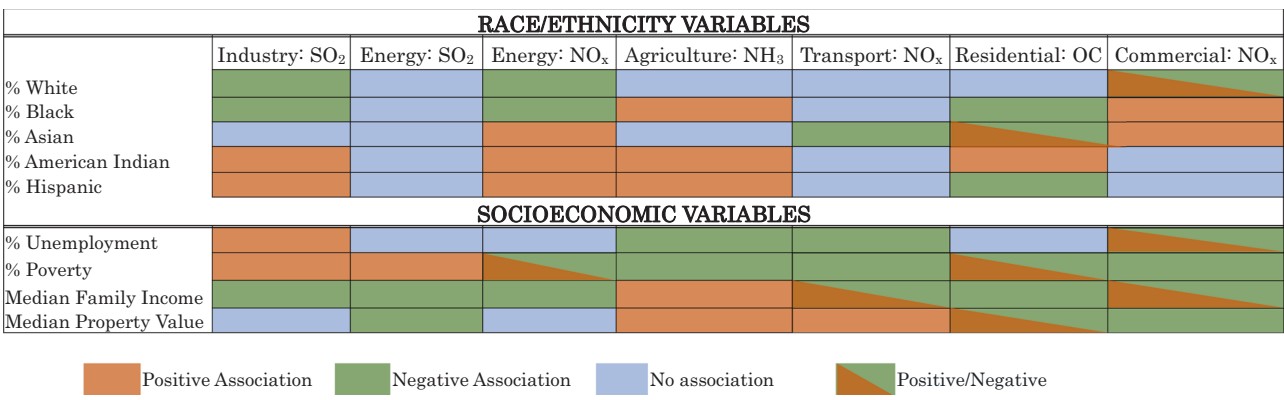

| | Industry: $SO_2$ | Energy: $SO_2$ | Energy: $NO_x$ | Agriculture: $NH_3$ | Transport: $NO_x$ | Residential: OC | Commercial: $NO_x$ |
|---|---|---|---|---|---|---|---|
| **RACE/ETHNICITY VARIABLES** | | | | | | | |
| % White | | | | | | | |
| % Black | | | | | | | |
| % Asian | | | | | | | |
| % American Indian | | | | | | | |
| % Hispanic | | | | | | | |
| **SOCIOECONOMIC VARIABLES** | | | | | | | |
| % Unemployment | | | | | | | |
| % Poverty | | | | | | | |
| Median Family Income | | | | | | | |
| Median Property Value | | | | | | | |

Positive Association   Negative Association   No association   Positive/Negative

**Fig. 4 | Summary of results categorizing the associations into four groups.** (1) Positive association (orange): a unit increase in the demographic variable is associated with a percentage point increase in the relative emissions change (e.g., a larger relative emissions increase or a smaller relative reduction); (2) Negative associations (green): a unit increase in the demographic variable is associated with a percentage point decrease in the relative emissions change (e.g., a smaller relative emissions increase or larger relative reduction); (3) No association (blue) indicates no association between the demographic variable and the decennial relative emission change; and (4) Positive/Negative association (bi-color orange and green): a positive association in the lower values of the demographic variable and a negative in the higher values. This is an oversimplification of results. For detailed results, see Figs. 2 and 3.

emission reductions in energy generation at the lowest financial cost[23]. Despite its unarguable success in reducing net air pollutant emissions[24,25], there are equity concerns over using market-based strategies[26]. Previous studies have found that energy facilities are disproportionally located in low-income and communities of color[27–29]. Our study shows that from 1970 to 2010, an increase in the percentage of American Indian, Asian, or Hispanic population resulted in an increase in the energy $NO_x$ relative emissions change, whereas the opposite occurred with White and Black population percentages. For instance, an increase in the Hispanic population percentage from the national average of 4.4% to 75% resulted in a 50 pp increase in the relative change of energy $NO_x$ emissions; and a decrease in county White percentage from the national average of 87% to 25% led to 12.5 pp increase in the relative emissions change. Our results also show that an increase in median family income resulted in a decrease in both energy $NO_x$ and $SO_2$ relative emissions (e.g., an increase in median family income from the national average of $49 K to $100 K resulted in nearly 100 pp decrease in the energy $SO_2$ relative emissions change). In addition to other factors, such as historical racism in land distribution[30–32], our results suggest inequities in emissions reductions could contribute to persistent air pollution inequities. Thus, it is important to evaluate the effects of air quality policy on mitigating or aggravating racial/ethnic and socioeconomic disparities in air pollution exposure in addition to setting net emissions reduction targets[33].

Unlike most other pollutants we analyzed, agricultural $NH_3$ emissions increased on average from 1970 to 2010, and our findings suggest racial/ethnic disparities in the emissions changes. We found that an increase in county-level percent Hispanic and American Indian populations led to an increase in the relative change of $NH_3$ emissions. For instance, an increase in county Hispanic population percentage from the national average of 4.4% to 75% resulted in a 2.5 pp increase in the $NH_3$ relative emissions change. These findings remained consistent after accounting for county-level SES. However, we found the opposite association between percent poverty and unemployment with the relative change in agricultural $NH_3$ emissions, potentially indicating the importance of agriculture as a job source. Although the pp change in relative emissions as a result of an increase in county percent Hispanic or American Indian population was small compared to the effect we observed in the energy source sector, as emissions from other air pollutants are reduced, $NH_3$ becomes a critical contributor to the formation of harmful fine particulate matter ($PM_{2.5}$)[34,35]. Agriculture is the leading sector in the Eastern US in driving anthropogenic $PM_{2.5}$ pollution[34,36]. Although the CAA permits federal authorities to regulate

$NH_3$, no federal regulations or incentive programs require reductions in $NH_3$ emissions[37,38].

County average emissions of OC from the residential sector were low relative to other pollutants we analyzed. On average, the OC emissions changes were minor but with a slight upward trend, and our results suggest potential differences by socioeconomic status and race/ethnicity. For instance, an increase in county-level median family income from the national average of $50 K to $100 K led to a 5 pp decrease in the relative change of OC emissions, whereas an increase in the percent American Indian population from the national average of 1.3% to 50% resulted in 3 pp increase in the relative change of OC emissions. In North America, OC emissions from the residential sector are mainly from solid biofuel use for space heating[21]; however, only a tiny fraction of American households rely on solid biofuel—primarily wood[39]. Solid biofuel for space heating is more prevalent in counties that are predominantly American Indian[40], which aligns with our findings. Also, in 2014, the Energy Information Administration reported that from 2005 to 2012, Northeastern states saw at least a 50% increase in households that rely on wood as the primary heating fuel as heating oil prices rose[41]. Biofuels are often considered a renewable energy source and may be considered a better climate alternative to fossil fuels; however, biofuel combustion emits health-damaging air pollutants, including $PM_{2.5}$, affecting indoor air quality[42,43]—mainly when used in old heating appliances with limited emissions controls[42]. The CAA sets performance standards for residential wood heaters under the New Source Performance Standards. These standards help ensure new customers have access to cleaner burning models but do not apply to existing wood heaters in people's homes[44]. As we transition to clean renewable energy sources, it is crucial to incorporate energy equity programs and regulations that increase clean energy accessibility and minimize the number of households turning to less clean or safe heating options to limit financial stress[45]. Although the percentage of households using solid biofuels is low, our findings suggest it might encompass vulnerable communities.

Most of the associations between the demographic variables we analyzed and pollutant relative emissions changes were nonlinear, reflecting the complexity of these relationships. The nonlinear relationships between county-level socioeconomic characteristics and the relative $NO_x$ emissions changes from transportation particularly stood out. For instance, below a county-level median family income of $70 K, an increase in income was associated with an increase in the relative emissions change. However, for median family income above $70 K, an increase in income was associated with a decrease in the relative

emissions change. Other studies have found similarly shaped relationships between income measurements and air pollution emissions in the US and abroad[46,47]. These findings suggest that from 1970 to 2010, relative $NO_x$ emissions decreases have been more substantial in high-income counties. The overall reduction in $NO_x$ emissions from transportation has resulted from the many emissions regulations, standards, and routine vehicle inspection and maintenance programs established through the CAA. These regulations have pushed the development and implementation of newer technologies to make vehicles more fuel-efficient and their emissions cleaner[48]. However, despite the success at the national level, several studies have shown that the current distribution of traffic-related pollution disproportionally impacts vulnerable communities[49–51]. Addressing inequities in traffic-related air pollution exposure, thus, may require targeted strategies. Mitigating traffic-related pollution in the most affected areas will be vital to mitigating current disparities and preventing their worsening from unequal adoption of electric vehicles[52].

Importantly, not all air pollutant emissions reductions disproportionally affected all vulnerable populations. In our analysis, socioeconomic disparities were more prevalent than racial/ethnic but the results varied by air pollutant, source, and demographic variable. In some air pollution source sectors, the reductions benefited some vulnerable populations. For example, an increase in county-level percent Black population was associated with a decrease in the relative emissions change in the industrial, energy, and residential air pollution source sectors. We also found a similar beneficial association for the Asian population in the transportation sector. In contrast, percent American Indian and Hispanic populations were primarily associated with an increase in the relative emissions change. And across all air pollution source sectors, except agriculture, an increase in median family income was linked with a decrease in relative emissions change.

Like all studies, this analysis comes with caveats. First and foremost, studies suggest that micro-scale (e.g., neighborhood-level) inequities in air pollution are common[49]. However, the air pollution emission estimates were only available in 0.5º × 0.5º grid resolution (~55 × 55 km), precluding a subcounty-level analysis. As a result, our analysis provides limited insight into the factors that drive emissions injustice at the local level. Importantly, emissions are different from air pollution concentrations in geospatial variation. The density of pollution sources spatially might not have as much variation in a hyper-local spatial resolution (e.g., census tracts or blocks) compared to county-level. This also means that spatial variations in air pollution emissions do not perfectly capture population air pollution exposure variations. In this study, we provide information about potential country-wide inequalities in the distribution of air pollution sources by leveraging emission data that can inform federal, state, and county-level regulations and supplement local-level analysis. The air pollution emissions are estimates and come with uncertainties, which vary across source sectors, time, and geography. In addition, we present aggregated results for the contiguous United States, and associations may have important sub-national variability that we may not have been able to capture. We ran regional models, but statistical power was limited for those analyses, particularly for racial/ethnic groups with low population counts, such as American Indian and Asian populations, which we could not analyze in the regional models. Also, the census racial/ethnic definitions changed throughout the study period. As a result, population growth or stagnation of a group may partially result from a broadening or narrowing of the census racial/ethnic definitions. This may be partly why, in our analyses, associations between racial/ethnic variables and air pollutant emissions changes were not as common. Furthermore, Hispanic breakdowns for the racial groups were unavailable for 1970; thus, the racial groups included both Hispanic and non-Hispanic populations. In this study, we evaluated one pollutant associated with each source sector (two in the case of energy), but a single source sector emits multiple pollutants and each

may have different trends. Also, it is important to note that people are exposed to ambient air pollution concentrations, and various factors influence individual-level exposures (e.g., meteorology, human activity, housing quality etc.) which we did not consider in this study. Our analyses also encompassed a large number of models, and issues related to multiple comparisons are of relevance. Lastly, although we evaluated six major air pollution source sectors, they do not account for all pollution sources in the US. Despite these limitations, we provide a look at longitudinal trends in air pollution emissions from six major sources following the 1970 CAA. The large number of results and identified non-linearities meant that we were unable to discuss each finding in detail.

## Methods

We conducted a longitudinal analysis to investigate the association between county-level demographics and subsequent changes in air pollution emissions across the contiguous US from 1970 to 2010. In 1970, the continental US had 3106 counties, which increased to 3109 by 2010. From the analysis, we excluded Yellowstone National Park County (average population = 130 individuals), which had a negative property value in the 1970 census data. We removed three counties (Broomfield, CO; Washabaugh, SD; and Nansemond, VA) that existed for only one year during the study period; thus, we could not estimate a change in emissions. We also removed five small islands and coastal counties for which we could not estimate emissions (Nantucket, MA; Mathews, VA; San Juan, WA; Door, WI; Poquoson, VA). In 1970, the Census did not collect race/ethnicity and ethnicity data for DC or property value for Adam's County, WI; thus, these two counties only contributed to the analysis from 1980 to 2010.

### Predictor variables

We used county-level data from the American Decennial Census for 1970, 1980, 1990, and 2000 to characterize race/ethnicity and socioeconomic conditions (NHGIS database[53]) in the contiguous US (excluding AK, HI, and Puerto Rico). We estimated counties' percent population for five race/ethnicity categories (as defined by the Census Bureau): White, Black, American Indian, Asian, and Hispanic. The racial/ethnic categories included Hispanic and non-Hispanic populations because ethnicity breakdowns for racial groups were unavailable for 1970. Therefore, the racial/ethnic percentages by county could sum to more than 100%. We characterized county-level SES using median family income (in dollars), median property value (in dollars), percent population living under the poverty threshold as defined by the Census Bureau, and unemployment (the proportion of over 15-year-olds unemployed out of the civilian workforce). The Census did not collect median family income and property value in 1970. Thus, for 1970, we estimated the mean county income by dividing the aggregated family income by the total number of families and the mean property value by dividing the aggregated property value by the total household number (the aggregated variables were available in the 1970 decennial census). We used the estimated 1970 mean family income and property value as proxies for the 1970 medians. We converted monetary variables to 2010 dollars to adjust for inflation using the Consumer Price Index Research Series[54], (e.g., to adjust 2000 dollars to 2010: 2000 dollars × (2010 cpi-u-rs/2000 cpi-u-rs).

### Outcomes of Interest

We examined six emission source sectors: industry, energy, agriculture, commercial, residential, and on-road transportation. Each source sector emits multiple air pollutants; we focused on chemical pollutants associated with each sector that have been previously linked to wide-scale air pollution: sulfur dioxide ($SO_2$) for industry; ammonia ($NH_3$) for agriculture; nitrogen oxides ($NO_x$) for commercial and transportation sectors; organic carbon (OC) for the residential

sector; and $NO_x$ and $SO_2$ for the energy sector. Table 4 lists specific air pollution sources and fuels associated with each sector.

We obtained the emissions data from the Community Emissions Data Global Burden of Disease Map (CED$_{GBD-MAP}$) emissions inventory[21]. This is a gridded ($0.5° × 0.5°$ or ~$55 × 55$ km) global bottom-up emission inventory from 1970 to 2017. In summary, CED$_{GBD-MAP}$ uses data from various emissions inventories (GAINS, SPEW, US NEI, EDGAR, etc.) and activity data (energy consumption, population, etc.) to calculate global emissions estimates for each chemical compound. For years without available emissions, default estimates are calculated from a linear interpolation and available activity data. Then, local and regional inventories are used to scale sectoral emissions to the national level[21]. We focused on the decennial years 1970, 1980, 1990, 2000, and 2010. We processed the emissions data for each pollutant to estimate county-level emission in kg/km²/day. We first created a dataset containing the geographical boundaries of US counties for each decade from 1970 to 2010[53] and applied an area-weighted interpolation technique[55] to obtain area-weighted county emission fluxes for each decennial year, pollutant, and pollution source sector of interest. Counties falling within emissions grid cells but whose centroids were outside the contiguous US (0.16% of counties in the dataset) were not included to avoid data issues on the emissions dataset's geographical boundaries (e.g., emissions grid cells partially over sea and land).

For each county, we calculated the relative change in emission fluxes from one decennial to the following expressed as a percentage (e.g., relative change = [(emissions in 1980 – emissions in 1970) / emissions in 1970] × 100). A negative relative change reflects a decrease in emissions over the ten years, whereas a positive relative change represents an increase. For analysis, we temporally matched the decennial demographic/socioeconomic data with the relative change in emissions in the following ten years, i.e., we matched 1990 demographic data to the relative decennial emissions change between 1990 and 2000. Estimating relative changes in energy and industrial emissions resulted in observations with high (i.e., emissions changed from a value close to zero to a large value) and infinite values (i.e., emissions changed from nonzero to zero), respectively. These extreme values affect the proper modeling of associations. Thus, we removed five counties with infinite values in the industry sector ($N = 15/12,437$) and excluded observations in the top 5 percentile in the energy sector ($N = 621/12,437$).

## Covariate information
We included in the analysis population density (number of people per km²), categorical variables for the Environment Protection Agency (EPA) geographic regions (Supplementary Fig. 6), and urbanicity (Supplementary Fig. 7) to adjust for potential confounding bias. The ten EPA regional offices are responsible for executing programs within the assigned states and territories, and these programs might differ across regions[56]. We combined regions 1–3, which cover the northeast from VA to ME, into a single group for a total of 8 regions. We used the 2013 National Center for Health Statistics Urban-Rural Classification Scheme for Counties[57] to categorize counties into three urbanicity groups: metropolitan (population ≥ 50,000), micropolitan (50,000 > population ≥ 10,000), and non-urban (population <10,000). We allowed the urbanicity status of counties to change throughout the study period depending on their population size.

## Main analysis
We evaluated the association between each predictor of interest (county-level percent population White, Black, American Indian, Asian, and Hispanic, percent poverty and unemployment, median family income and property value) and decennial relative change in emissions from agriculture $NH_3$, commercial and transportation $NO_x$, residential OC, energy $NO_x$ and $SO_2$, and industry $SO_2$. We built separate models for each predictor variable (race/ethnicity and socioeconomic variables) and air pollution source sector for a total of 48 models. All models were adjusted for population density, urbanicity, EPA region, and time using year as a categorical variable (1970, 1980, 1990, and 2000). The models with the predictor variables percent Black, Asian, and American Indian population were also adjusted for percent White population because the percent White population in a county may co-vary both with the Black, American Indian, or Asian population percentages and the emission changes (see Supplementary Table 2 for a summary of the models). We extracted the association estimates for the White population percentage from the models that included the variables percent Black and percent White. Other studies have used a similar approach[58,59]. We used the following hierarchical model formulation:

$$Y_{tc} = (\beta_0 + b_{0,c|s}) + \beta_x X_{tc} + \beta_z Z_{tc} + \varepsilon_{tc} \tag{1}$$

In the model (1), $c$ represents a county in state $s$. $Y_{tc}$ is the relative change in emissions for a specific air pollution source sector (e.g., the relative change in $NH_3$ emissions from the agriculture) in county $c$ from year $t$ to $t + 10$. We included random intercepts for counties nested within states ($b_{0,c|s}$), to account for within-county correlation in emissions changes over time. $X$ is the predictor of interest (demographic and socioeconomic factors) and the covariates are summarized in vector $Z$. In essence, with this model, we estimated the association between county-level demographics and the decennial relative emissions change. That is, per one unit increase in the demographic variable, what is the percentage point change in the decennial relative emissions? Please note that we used racial/ethnic and socioeconomic characteristics of counties in year $t$ to estimate the relative emissions change from year $t$ to $t + 10$. Because we did not have emissions data for the year 2020, the last year of demographic data we used is 2000 to

**Table 4 | Air pollutant fuel and related activities**

| Emission Source Sector | Air pollutant | Primary contributing fuel(s) | Example activities |
|---|---|---|---|
| Industry | $SO_2$ | Coal and oil | Industrial boilers in the production of iron-steel, cement, metals, etc. |
| Energy generation | $SO_2$ | Coal | Electricity production, fuel production and transportation, oil and gas fugitive/flaring, etc. |
| Energy generation | $NO_x$ | Coal, oil, and gas | |
| Agriculture | $NH_3$ | N/A | Manure management, soil emissions, enteric fermentation, etc. |
| Transportation | $NO_x$ | Oil and gas | On-road transportation |
| Residential energy combustion | OC | Solid biofuel | Cooking, space heating, residential waste burning |
| Commercial combustion | $NO_x$ | Oil and gas | Combustion in service-providing facilities (e.g., religious facilities, local, state, and federal government, institutional living quarters, sewage treatment facilities, restaurants, and more) |

Primary fuel type contribution to emissions of each air pollutant per source sector and sample activities. For more see McDuffie et al.[21].

evaluate the relative decennial emissions change that occurred from 2000 to 2010. Thus, the data included four time points (1970, 1980, 1990, and 2000) and 12,437 observations total (~3000 counties per time point).

We evaluated nonlinearities in the predictors of interest and the continuous covariate (population density) to avoid potential model misspecification and comprehensively characterize the associations with the outcome variable[60]. We used penalized splines to flexibly model the associations and the generalized cross-validation (GCV) criterion to determine whether an association deviated from linearity (estimated degrees of freedom [edf] > 1). If relationships were non-linear, we included the independent variable of interest (racial/ethnic and socioeconomic variables) with a natural spline (4 degrees of freedom, using the predictor's distribution mean as the reference) in the model. In the final models, we used natural splines rather than penalized splines for the independent variables of interest, as penalized splines can be too flexible and sensitive to influential observations. If we found no evidence of nonlinearity (edf = 1), we included the independent variable as a continuous measure and modeled its linear association with the dependent variable.

### Sensitivity analyses
It is often the case that racial/ethnic variables are correlated with socioeconomic variables. Thus, we also evaluated the robustness of our results for the racial/ethnic variables of interest to confounding by SES. We modeled the association between race/ethnicity and the outcome adjusting for socioeconomic variables (family income, poverty, unemployment, property value) in addition to population density, urbanicity, EPA region, and the categorical variable year (see models summary Supplementary Table 3). This analysis followed the structure of Model 1 but with the additional socioeconomic variables included as covariates. We modeled all nonlinear associations with penalized splines.

### Secondary analyses
The associations between demographic and socioeconomic factors and the relative decennial change in emissions may vary across regions. However, regional sub-analyses for all emissions source sectors and sociodemographic variables would substantially increase the number of models and the chance of detecting false positive associations. We, therefore, a priori decided on a subset of regional sub-analyses. Specifically, we modeled the associations between median family income and percent population Black and White with industry $SO_2$ and transportation $NO_x$, separately in each EPA region. We excluded the rest of the variables to make the regional analysis computationally feasible and also because the percentage for the other racial/ethnic groups (American Indian, Asian, and Hispanic) were very low in some regions of the US; thus, we lack the power to model their associations regionally accurately. We performed the regional analysis using Model 1 and penalized splines for all models. The regional analyses included 32 separate models (Supplementary Table 4).

We report association curves with 95% confidence intervals (CI) for all relationships and effect estimates with 95% CI only for the linear associations. Linear effect estimates are expressed as the percentage point (pp) increase/decrease in the relative emission change per 10 pp increase in the demographic variable of interest.

### Reporting summary
Further information on research design is available in the Nature Portfolio Reporting Summary linked to this article.

## Data availability
The raw emissions data used in this study can be accessed at https://zenodo.org/record/3754964#.Y-bEE-zMLOS and the raw demographic data via https://data.census.gov/ or NHGIS.org. The processed emissions and demographic data have been deposited in a publicly available GitHub repository [https://github.com/yanellinunez/USA_emissions_code].

## Code availability
All analyses were performed using R version 4.1.2 running under macOS Monterey 12.3. Primarily, the package tidyverse version 2.0.0 was used for the cleaning and wrangling of demographic data; package RNetCDF version 2.8-1 and PCICt version 0.5-4.4 for the procession of emissions data; function gamm4 from the package gamm4 version 0.2-6 was used for all modeling analyses; and ggplot2 version 3.4.2 was used for data visualizations. For a complete list of the packages and code used in the analysis, see https://github.com/yanellinunez/USA_emissions_code[61].

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

## Acknowledgements

We thank the National Institute of Environmental Health Sciences for its funding, which made this work possible (P30 ES009089, R01 ES030616 [M.-A.K.], R01 ES028805 [M.-A.K.], and T32 ES007322), and The Thomas F. and Kate Miller Jeffress Memorial Trust, Bank of America, Trustee (L.R.F.H.). We also thank the Health Effects Institute (HEI, R-82811201 [L.R.F.H.]; research described in this article was conducted under contract to the HEI an organization jointly funded by the United States EPA and certain motor vehicle and engine manufacturers. The contents of this article do not necessarily reflect the views of HEI, or its sponsors, nor do they necessarily reflect the views and policies of the EPA or motor vehicle and engine manufacturers).

## Author contributions

The authors Y.N., M.-A.K., J.A.C., J.G., and L.R.F.H. conceptualized the research questions. Y.N., J.G., and M.-A.K. identified the most appropriate statistical methods to address the research question at hand. Y.N., J.B., J.S., and M.D. conducted the analysis, and Y.N. created all visualizations. M.-A.K. supervised all components of the analysis. Y.N. wrote the first draft of the manuscript, and M.-A.K., J.A.C., J.G., L.R.F.H., E.M.K., J.B., J.S., M.D., and E.M. reviewed the draft and provided feedback.

## Competing interests

The authors declare no competing interests.
