## [Peer Review File · Nature Communications]

An Environmental Justice Analysis of Air Pollution Emissions in the United States from 1970 to 2010REVIEWER COMMENTS

Reviewer #1 (Remarks to the Author):

The authors examined temporal trends in air pollution emissions from various sources from 1970 to 2010 across the US across strata of various county-level racial and sociodemographic characteristics. The focus on emissions in this paper is a nice addition to the current environmental justice literature related to air pollution. The paper is clearly written, and the Figures are very nice. The authors may want to consider the points below in revising their paper. All of these points are minor.

Comments

- It might be helpful to show the correlation between emissions and concentrations (if possible). In my experience, this correlation can sometimes be weak, presumably because in some cases emissions are released through stacks that are design to disperse the pollutants over some distance and thus the emission may not contribute substantially to the concentration in the immediate area of its release. It might also be interesting to know if the correlation between emissions and outdoor concentrations also varies by racial/sociodemographic characteristics as this would inform possible interventions. This will depend on the spatial scale of course, and if the authors think that the resolution of their emissions predictions would be useful for such an analysis. Just a thought.

- Are there spatial variations in the quality of the emissions inventories or is this stable across the US?

Reviewer #2 (Remarks to the Author):

Review of "Air Pollution Emissions Trends in the Contiguous United States from 1970 to 2010: An Environmental Justice Analysis" for Nature Communications (NCOMMS-23-09556).

This paper describes statistical analysis of air pollution emission trends and demographic variables in the continental US from 1970-2010. Emissions data are from a global inventory and demographic data are from the census. They report results as association in which different groups either experience positive or negative associations with changes in emissions. There is a lot of heterogeneity across the demographic variables.

This is an important topic given the need to address environmental justice challenges around air pollution. I found the paper interesting but had a hard time figuring out the important conclusions and their implications with respect to policy. There were several weaknesses and (for me) points of confusion in the paper.

First the analysis is done at the county level while most air pollution injustice occurs at a much finer spatial scale. This weakness is acknowledged in the paper, but means that the paper likely provides limited insight into the factors that are driving environmental injustice.

The paper examines four pollutants (NO_x, SO₂, NH₃, and OC). The primary impact of three of these pollutants (NO_x, SO₂, and NH₃) is as precursors for secondary PM_{2.5} formation. Therefore, emissions of these pollutants likely have only a relatively modest effect on near-field (within the county) air pollution and environmental injustice. While NO_x (NO₂) has important pollutant, primary PM_{2.5} and air toxics likely are more important but unfortunately were not included in the analysis. Therefore it is not clear emission trends of the studied pollutants are of great interest from an environmental justice perspective.

These pollutants are governed by national policies (or no policies in the case of NH₃). What are the implications of that in terms of intent. The location of the emission reductions are also largely determined by the location of the facility versus the policy. What are the implications of this with respect to the work? It seems like the bigger issue is the original siting decisions versus application of emission control policies.

I have questions about the discussion of the role of cap-and-trade policy from the acid rain program since the majority of the data are from before that policy went into effect. While the statement “there are equity concerns over using market-based strategies” is true I am not convinced “our results align with such concerns.” Figure 2 shows that energy SO₂ (the primary target of acid rain program) are not statistically significant for any demographic. For energy NO_x, associations are negative for black but positive for all other categories (including white). Is that alignment with the concerns.

I was confused about interpreting the data. My understanding is that positive associations are bad (i.e. changes are worse than national average) while negative numbers are good (changes in emissions greater than national averages). Is that correct? I also assume that associations are relative to the national averages? These needed to be more clearly defined.

I was confused by Figure 4. It suggests there are negative associations Industry SO₂ with white population, but Figure 2 indicates they are positive. Same comment for energy NO_x and white. I did not systematically compare all categories but this added to my confusion about trying to interpret the results. Maybe I am missing something.

“However, neither modeled nor measured concentrations provide information about the air pollution sources contributing to the observed disparities.” While I have not seen this sort of analysis, a paper by Chris Tessum (I believe reference 7) does link exposure disparities back to sources using a model (others who have done this at regional scales).

Reviewer #3 (Remarks to the Author):

Nunez et al calculated the relative changes in emissions for each county across the US from 1970 - 2010. They then combined the relative emissions changes with the county-level demographic data, and estimated the associations between the demographic variables and the relative emission changes. They found non-linear associations for many sector/demographic pairs and documented relative differences in the emission trends.

Overall, I think this paper addresses a very important topic and the paper will potentially generate impacts on policy discussions. The researchers have compiled a great dataset from multiple sources over a long time, which is admirable. However, I think the paper could be improved by being much clearer about the interpretation and presentation of their main results. In particular, I think an expanded discussion of the relative emission changes (especially why is it an important metric to look at) should be included in the introduction and discussion section. Below, I have two major comments and several minor ones. I hope my comment will help improve this important paper.

Major:

1. One of the biggest challenges I have in reading this paper is how to interpret the associations between relative emission changes and demographic variables. Compared to changes in absolute emissions across different groups, the relative metric is more difficult to interpret (at least for me) in the context of EJ. I think it will benefit from more discussion, especially at the beginning of the paper, about why is a useful metric and how we should think about it (in relation to the broader discussion of EJ).

One major thing that stood out for me is -- I wonder if the association between racial/SES variables and relative emission changes (the metric being estimated in the paper) needs to be considered together with the associations between racial/SES variables and the absolute emissions. For example, if one sector's emission was concentrated in one racial group, then what should an "equitable emission reduction strategy" look like? Should we target equal relative reduction (e.g., a reduction by 20% across all counties), OR should we target higher-than-proportional reduction from the high-emitting counties, OR should we target equal absolute reductions (e.g., a reduction of 100 tons) which would then result in a lower-than-proportional reduction for those highly-burden communities? Because of the conceptual challenges above, I find it difficult to understand the associations estimated in the paper.

I would suggest 1) a more expanded discussion on the above conceptual challenges in the introduction and the discussion section, and 2) for each panel in figures 2 and 3, show the results of the association between the SES variables and the absolute emissions (perhaps as SI figures).

2. Partly related to the comment above, I find it challenging to fully understand Figures 2 and 3. The shape of the association curve is clear-- whether it's linear or not, and the overall directional relationship. But I don't know how to interpret the magnitude of each point on the curve. The figure is hard to interpret partly because of the narrow support of the percentage of each demo group (as shown in the top panel). Would it make more sense to just plot the curve over the range that has the most data coverage for each racial/SES group?

Another important question about this figure is the interpretation of the y-axis. The caption suggests that the y-axis just shows the relative emission changes. If that's the case, are the magnitudes comparable across each row (for the same sector)? Should the range of the y-axis be identical in the same row? For example, in the second row of figure 2 (industry SO₂ emissions), it seems like the first panel suggests that SO₂ emissions decreased in white-majority counties, but then increased in zero-black counties. I would encourage the authors to examine each panel's normalization baselines (i.e. the y=0 point) and make them comparable and easy to interpret when possible.

Minor:

1. The abstract can be revised to clarify the method and main results. While the expression of "association" is commonly seen in the public health literature, I think a more direct and plain-language description may be more appropriate in the abstract. For example, it could be something like "counties with higher Hispanic population percentages experience a smaller decline in SO₂ emissions from the energy sector".

2. It will be helpful to briefly explain the method and estimated metrics in the abstract after the first sentence. For example, it could be something like “Here, we calculate the relative emission changes for each county and examine the differential changes across counties with xxx”

3. As the emissions inventory is essential for this paper’s method, I suggest including more descriptions of the CED emissions data. Despite being a popular data source, it might not be clear to everyone what it is. Is this mostly a compilation of NEI and local emissions inventory for the US? This should be in both the methods section and the last paragraph of the introduction section.

4. It is unclear why NO_x is selected as the precursor of the commercial sector. A brief explanation in the methods section (including what are the main emission activities in the commercial sector) would be helpful.

5. The results section starts with a discussion of the changes in the demographic structure. This makes me wonder about the role of demographic changes in the association. It would be interesting to see simple sensitivity analyses that calculate the association with fixed demographic patterns. This would then be helpful to understand how demographic changes influence the EJ pattern.

REVIEWER COMMENTS

Reviewer #1 (Remarks to the Author):

The authors examined temporal trends in air pollution emissions from various sources from 1970 to 2010 across the US across strata of various county-level racial and sociodemographic characteristics. The focus on emissions in this paper is a nice addition to the current environmental justice literature related to air pollution. The paper is clearly written, and the Figures are very nice. The authors may want to consider the points below in revising their paper. All of these points are minor.

Comments

1. It might be helpful to show the correlation between emissions and concentrations (if possible). In my experience, this correlation can sometimes be weak, presumably because in some cases emissions are released through stacks that are design to disperse the pollutants over some distance and thus the emission may not contribute substantially to the concentration in the immediate area of its release. It might also be interesting to know if the correlation between emissions and outdoor concentrations also varies by racial/sociodemographic characteristics as this would inform possible interventions. This will depend on the spatial scale of course, and if the authors think that the resolution of their emissions predictions would be useful for such an analysis. Just a thought.

Thank you for your comment and insight. In this study, we are particularly interested in emissions as those are related to specific sources. Other papers have evaluated inequities in $PM_{2.5}$ air concentrations; the limitation of focusing on air pollution concentrations is that they are not directly linked to a source that can be targeted for policy or regulation. We specifically wanted to evaluate emissions to provide insight into potential air pollution sources contributing to inequities.

We agreed that evaluating the correlation between air pollution emissions and concentrations is important, but we believe it is outside the focus of this study. As you mentioned, correlations between emissions and air pollution concentrations vary by pollutant and likely also by time and space. Additionally, as you know, many other factors influence the correlation between air pollution concentrations and emissions, including weather, air patterns, landscape, etc. Given this complexity, evaluating the correlation between pollution concentrations and

emissions can be a publication on its own. Furthermore, multiple limitations hinder our ability to evaluate the correlation between air pollution concentrations and emissions during our study period; the main one is that we don't have access to historical air pollution concentration data dating back to 1970.

2. Are there spatial variations in the quality of the emissions inventories or is this stable across the US?
 - a. Add in the limitations that it is possible there is spatial and temporal variation in the emissions

Thank you for your comment. We do not anticipate the quality of emissions inventories to greatly varied within the US, but spatial variation in data quality can't be discarded. Erin E. McDuffie et al.,¹ also mentioned that temporal variation in the quality of inventories and variation in the quality of inventories across sectors are significant contributors to CED_{GBD-MAPS} data uncertainties. We have added a sentence in the limitations stating the various potential sources of data uncertainty.

"The air pollution emissions are estimates and come with uncertainties, which vary across sectors, time, and geography."

Reviewer #2 (Remarks to the Author):

Review of "Air Pollution Emissions Trends in the Contiguous United States from 1970 to 2010: An Environmental Justice Analysis" for Nature Communications (NCOMMS-23-09556).

This paper describes statistical analysis of air pollution emission trends and demographic variables in the continental US from 1970-2010. Emissions data are from a global inventory and demographic data are from the census. They report results as association in which different groups either experience positive or negative associations with changes in emissions. There is a lot of heterogeneity across the demographic variables.

This is an important topic given the need to address environmental justice challenges around air pollution. I found the paper interesting but had a hard time

¹ <https://essd.copernicus.org/articles/12/3413/2020/>

figuring out the important conclusions and their implications with respect to policy. There were several weakness and (for me) points of confusion in the paper.

1. First the analysis is done at the county level while most air pollution injustice occurs at a much finer spatial scale. This weakness is acknowledged in the paper, but means that the paper likely provides limited insight into the factors that are driving environmental injustice.

Thank you for your comment; we understand your concern. However, in this study, we are focusing on air pollution emissions instead of concentrations. Emissions are different than concentrations in the spatial variation and density of sources; specifically, it is likely that the distribution of air pollution sources has less variability at higher spatial resolutions than air pollution concentrations do. We acknowledge that our study does not provide insight into inequities that occur on a finer spatial scale, like neighborhood or census block; however, our analysis provides insight into macro-level inequalities nationwide, which is also valuable information for federal and state regulations. States can use counties as units to operate or provide funding to, which would be more complicated in a census tract or block scale. Inequities exist in multiple layers and do not only occur at the neighborhood level. Our study and other studies^{2,3,4} show that higher-level inequities also exist.

We have added more information on the limitations of our spatial resolution at the end of the Discussion section:

“Like all studies, this analysis comes with caveats. First and foremost, studies suggest that micro-scale (e.g., neighborhood-level) inequities in air pollution are common⁵¹. However, the air pollution emission estimates were only available in 0.5o 0.5o grid resolution (~ 55 x 55 km), precluding a subcounty-level analysis. As a result, our analysis provides limited insight into the factors that drive emissions injustice at the local level. Importantly, emissions are different from air pollution concentrations in geospatial variation. The density of pollution sources spatially might not have as much variation in a hyperlocal spatial resolution (e.g., census tracts or blocks) compared to county-level. In this study, we provide information about potential country-wide inequalities in the distribution of air pollution sources

² <https://iopscience.iop.org/article/10.1088/1748-9326/abe4f7/meta>

³ <https://www.nature.com/articles/s41467-023-38084-6>

⁴ <https://www.nature.com/articles/s41370-022-00462-5>

by leveraging emission data that can inform federal, state, and county-level regulations and supplement local-level analysis.”

2. The paper examines four pollutants (NO_x, SO₂, NH₃, and OC). The primary impact of three of these pollutants (NO_x, SO₂, and NH₃) is as precursors for secondary PM_{2.5} formation. Therefore, emissions of these pollutants likely have only a relatively modest effect on near-field (within the county) air pollution and environmental injustice. While NO_x (NO₂) has important pollutant, primary PM_{2.5} and air toxics likely are more important but unfortunately were not included in the analysis. Therefore it is not clear emission trends of the studied pollutants are of great interest from an environmental injustice perspective.

Thank you for your comment. We understand your concern. Unfortunately, the CED_{GBD-MAP} inventory does not include primary PM_{2.5} emissions data, so we can't analyze emissions of this pollutant in this study. The inventory does include emissions of volatile organic compounds (VOC), but given the current density of our paper, we believe including an additional pollutant would make our manuscript even more challenging to digest. The primary focus on our paper is on air pollutants emissions, thus, we approached our pollutant selection by first identifying major air pollution sources, then the pollutant compound more generally associated with each source.

Correct NO_x, SO₂, and NH₃ are precursors for PM_{2.5}. Importantly, to reduce PM_{2.5} air pollution concentrations, emissions of PM_{2.5} precursors need to be addressed. In this paper, we focused on emissions of, among others, PM_{2.5} precursors to provide insight into specific targetable sources. As you mentioned, some of these pollutants may only have a modest impact on nearby communities because they readily disseminate in the air. However, even if it is a modest impact, they have an impact nonetheless, which, if many sources in the area, eventually add up. Thus, many communities are pushing and have pushed for environmental legislature that considers the cumulative impact of environmental exposures.^{5,6,7} Additionally, the

⁵ <https://bloustein.rutgers.edu/the-backdrop-of-newarks-environmental-justice-and-cumulative-impacts-ordinance/>

⁶ <https://www.complexeffects.com/p/albuquerque-residents-seek-environmental>

⁷ <https://enviro.blr.com/environmental-news/EHS-management/EPA-and-multistate-environmental-law-regulations/New-Yorks-cumulative-impacts->

pollutants we evaluated are emitted along with other air pollutants, many of which are harmful to health. Previous studies have shown the adverse health impacts pollution sources have on fence-line communities.⁸ Although PM_{2.5} precursors not only affect the county where the source is located, we believe the impact of the emissions in nearby communities is worth considering and studying.

3. These pollutants are governed by national policies (or no policies in the case of NH₃). What are the implications of that in terms of intent. The location of the emission reductions are also largely determined by the location of the facility versus the policy. What are the implications of this with respect to the work? It seems like the bigger issue is the original siting decisions versus application of emission control policies.

Thank you for the comment. Our intent is to provide insight into nationwide patterns in air pollution emissions changes. Our study shows that air pollution emissions changes from 1970 to 2010 varied by demographic group for some sectors. We agree that the original siting of facilities is a big issue and has contributed to long-term environmental inequities. However, there are thousands of existing sites from which emissions can be reduced and sites that could be discontinued. I apologize if I'm misunderstanding your comment, but it is my understanding that national policies apply to all states. All states are required to submit to the EPA an air quality State Implementation Plan outlining state plans for meeting NAAQS.⁹ Furthermore, air pollution regulations not only apply to emissions but also affect the siting location of new sites or the technology existing facilities are required to use in order to reduce emissions.

4. I have questions about the discussion of the role of cap-and-trade policy from the acid rain program since the majority of the data are from before that policy went into effect. While the statement "there are equity concerns over using market-based strategies" is true I am not convinced "our results align with such concerns." Figure 2 shows that energy SO₂ (the primary target of

law#:~:text=%E2%80%9CThe%20cumulative%20impacts%20law%20also,New%20York%20Law%20Journal%20continues.

⁸ <https://link.springer.com/article/10.1007/s40572-020-00263-8>

⁹ <https://www.epa.gov/air-quality-implementation-plans/how-epa-works-states-sips#:~:text=States'%20role%3A,the%20Governor's%20designee%20to%20EPA.>

acid rain program) are not statistically significant for any demographic. For energy NO_x, associations are negative for black but positive for all other categories (including white). Is that alignment with the concerns.

Thank you for your comments. Phase one of the Acid Rain Program (ARP) began in 1995 for SO₂ and 1996 for NO_x. The years by which to accomplish the targeted emission reductions were 2000 and 2010 for SO₂ and NO_x, respectively.¹⁰ Our analysis uses emissions data from 1970 to 2010, which means it captures the first 15 years or so of the ARP, including up to the target years for accomplishing the emissions reductions. However, we have softened the language describing the relevance of our results since, in this analysis, we are not evaluating any particular policy. You are correct that we should not draw such strong conclusions.

Regarding the results shown in Figure 2, you are correct. There are no associations between emissions changes in energy SO₂ and any of the race/ethnicity variables. You are right that NO_x associations are negative for Blacks and positive for Asians, American Indians, and Hispanics. However, the association with White is not positive; it is negative. The curve has a negative slope. A unit increase in the White population percentage is associated with a decrease in the relative change of energy NO_x emissions. In Figures 2 and 3 the association curves are plotted in reference to the independent variable mean. So for example, a change in the county-level White percentage from the national mean of 87% to 50% (37 percentage points [pp] *decrease*) results in 5 pp *increase* in the relative emissions change; and a change in county-level White percentage from 87% to 25% leads to about 12.5 pp increase in the relative emissions change. In Figure 3, we also see a negative association between median family income and the relative emissions change in energy NO_x and SO₂, and a positive association between county-level percent poverty and the relative emissions change in energy SO₂. For example, an increase in percent poverty from the nationwide average of 16.5% to 40% results in a 100 pp increase in the energy SO₂ relative emissions change. This is why we have written that our results align with equity concerns over market-based pollution reduction strategies.

In addition to editing the paragraph discussing the Acid Rain Program, we also edited the legends of Figures 2-3 to more clearly describe how to interpret the association curves. See our response to the next comment.

¹⁰ <https://www.epa.gov/acidrain/acid-rain-program#:~:text=The%20SO2%20program%20sets,the%20power%20sector%20in%201980.>

“Despite its unarguable success in reducing net air pollutant emissions 24,25, there are equity concerns over using market-based strategies 26. Previous studies have found that energy facilities are disproportionately located in low-income and communities of color 27-29. Our study shows that from 1970 to 2010, an increase in the percentage of American Indian, Asian, or Hispanic population resulted in an increase in the energy NOx relative emissions change, whereas the opposite occurred with White and Black population percentages. For instance, an increase in the Hispanic percentage average from the national average of 4.4% to 75% resulted in a 50 pp increase in the relative change of energy NOx emissions; and a decrease in county White percentage from the national average of 87% to 25% led to 12.5 pp increase in the relative emissions change. Our results also show that an increase in median family income resulted in a decrease in both energy NOx and SO2 relative emissions (e.g., an increase in median family income from the national average of \$49K to \$100K resulted in nearly 100 pp decrease in the energy SO2 relative emissions change). In addition to other factors, such as historical racism in land distribution³⁰⁻³², our results suggest inequities in emissions reductions could contribute to persistent air pollution inequities. Thus, it is important to evaluate the effects of air quality policy on mitigating or aggravating racial and socioeconomic disparities in air pollution exposure in addition to setting net emissions reduction targets ³³.”

5. I was confused about interpreting the data. My understanding is that positive associations are bad (i.e. changes are worse than national average) while negative numbers are good (changes in emissions greater than national averages). Is that correct? I also assume that association are relative to the national averages? These needed to be more clearly defined.

We apologize for the confusion. You are correct about positive associations being “bad” and negative associations being “good.” In Figures 2-3, the y-axis is the percentage point (pp) change in the relative emissions (we have corrected the labels) as the demographic value changes from the national average to another value (e.g., an increase from the county Hispanic percentage average of 4.4% to 75%, results in

the energy NO_x relative emissions change increasing 50 percentage points). In simple terms, our models are regressions where the independent variable is the demographic variable, and the outcome is the decennial relative emissions change. The associations, thus, indicate the percentage point change in the relative emissions (dependent variable) per unit increase in the demographic variable (independent variable). We edited the paragraph describing the main analysis model in the methods section and the legend in Figures 2 and 3, providing more clear information on how to interpret the results.

METHODS SECTION:

“...In the above model, c represents a county in state s . $Y_{t,c}$ is the relative change in emissions for a specific sector (e.g., the relative change in NH₃ emissions from the agriculture sector) in county c from year t to $t + 10$. We included random intercepts for counties nested within states ($b_{0,c/s}$), to account for within-county correlation in emissions changes over time. X is the predictor of interest (demographic and socioeconomic factors) and the covariates are summarized in vector Z . In essence, with this model (and all subsequent models), we estimated the association between county-level demographics and the decennial relative emissions change. That is, per one unit increase in the demographic variable, what is the percentage point change in the decennial relative emissions? Please note that we used racial/ethnic and socioeconomic characteristics of counties in year t to estimate the relative emissions change from year t to $t + 10$. Because we did not have emissions data for the year 2020, the last year of demographic data we used is 2000 to evaluate the relative decennial emissions change that occurred from 2000 to 2010. Thus, the data included four time points (1970, 1980, 1990, and 2000) and 12,437 observations total (~ 3000 counties per time point).”

LEGENDS:

“Figure 2: Association curves for the racial/ethnic groups. All models are adjusted for population density, urbanicity, EPA geographic region, and year.

The x-axis was scaled to improve the visibility of the curve in the low/high x values (squared root for percent White and squared for the other demographics). The curves represent the relationship between the demographic variables and change in the relative emissions (percentage point difference) relative to the mean of the demographic variable (e.g., from the national Hispanic average of 4.5% [during the study period] to 75%, energy NOx relative emissions increase 50 percentage points). The shaded area on the curve is the 95% confidence interval. Linear associations are indicated by the “Linear” label. Density plots for the independent variables are shown at the top of each column.”

“Figure 3: Association curves for the socioeconomic variables. All models are adjusted for population density, urbanicity, EPA geographic region, and year. The curves represent the relationship between the demographic variables and change in the relative emissions (percentage point difference) relative to the mean of the demographic variable (e.g., from the national poverty average of 16.5% [during the study period] to 40%, energy SO2 relative emissions increase 100 percentage points). The shaded area on the curve is the 95% confidence interval. All associations were nonlinear. Density plots for the independent variables are shown at the top of each column.”

6. It suggests there are negative associations Industry SO2 with white population, but Figure 2 indicates they are positive. Same comment for energy NOx and white. I did not systematically compare all categories I was confused by Figure 4. but this added to my confusion about trying to interpret the results. Maybe I am missing something.

We apologize for the confusion. In summary, a negative/positive association is not based on whether the curve lies on the positive or negative side with respect to the y-axis values, but instead on whether the slope is positive or negative. An association with a negative slope is negative, and an association with a positive slope is positive. We added a short example in the legend of Figures 2 and 3, to help

the reader with the interpretation of the results. Please see our answer to comment 4. We also edited the legend of Figure 4, adding more details for the interpretation of the results.

“Figure 4. Summary of results categorizing the associations into four groups. 1) Positive association (orange): a unit increase in the demographic variable is associated with a percentage point increase in the relative emissions change (e.g., a larger relative emissions increase or a smaller relative reduction); 2) Negative associations (green): a unit increase in the demographic variable is associated with a percentage point decrease in the relative emissions change (e.g., a smaller relative emissions increase or larger relative reduction); 3) No association (blue) indicates no association between the demographic variable and the decennial relative emission change; and 4) Positive/Negative association (bi-color orange & green): a positive association in the lower values of the demographic variable and a negative in the higher values. This is an oversimplification of results. For detailed results, see Figures 2 and 3.”

1. “However, neither modeled nor measured concentrations provide information about the air pollution sources contributing to the observed disparities.”
While I have not seen this sort of analysis, a paper by Chris Tessum (I believe reference 7) does link exposure disparities back to sources using a model (others who have done this at regional scales).

Thank you for pointing this out; we have now revised our manuscript based on your comment. While we still compare the paper by Chris Tessum to our own analysis, we now highlight the difference in temporal scope of the two analyses.

“Most evidence on targetable air pollution sources comes from cross-sectional analyses. Several studies have used residential proximity to pollution sources as a metric to evaluate inequities. These studies have found racial and socioeconomic disparities in the spatial distribution of industrial facilities^{6,13}, landfills, hazardous waste sites^{14,15}, gas¹⁶ and coal-fired power plants¹⁷, roadways¹⁸, and other pollution sources^{19,20}. Residential proximity studies are generally cross-sectional and often focus on a single air pollution sector. Other studies have evaluated inequity in

multiple air pollution sources by leveraging data from local emissions inventories, but these analyses also focused on a single time point⁷. Thus, they do not provide information about temporal inequity trends in emissions changes.”

Reviewer #3 (Remarks to the Author):

Nunez et al calculated the relative changes in emissions for each county across the US from 1970 - 2010. They then combined the relative emissions changes with the county-level demographic data, and estimated the associations between the demographic variables and the relative emission changes. They found non-linear associations for many sector/demographic pairs and documented relative differences in the emission trends.

Overall, I think this paper addresses a very important topic and the paper will potentially generate impacts on policy discussions. The researchers have compiled a great dataset from multiple sources over a long time, which is admirable. However, I think the paper could be improved by being much clearer about the interpretation and presentation of their main results. In particular, I think an expanded discussion of the relative emission changes (especially why is it an important metric to look at) should be included in the introduction and discussion section. Below, I have two major comments and several minor ones. I hope my comment will help improve this important paper.

Major:

1. One of the biggest challenges I have in reading this paper is how to interpret the associations between relative emission changes and demographic variables. Compared to changes in absolute emissions across different groups, the relative metric is more difficult to interpret (at least for me) in the context of EJ. I think it will benefit from more discussion, especially at the beginning of the paper, about why is a useful metric and how we should think about it (in relation to the broader discussion of EJ).

Thank you for your comment. There are arguments to be made for using the relative emissions change as well as for using the absolute change. We agreed that interpreting relative emissions changes can be more difficult, especially when used as outcomes in a regression model as we did. However, we strongly believe that in order for emissions reductions to be equitable, places with higher emissions should

have higher reductions (others have made similar recommendations [e.g., Yuzhou Wang et al.]). For example, if in two different counties, NH₃ emissions decreased by 20 tons, and we are evaluating absolute changes, then the reduction would be equitable even if the initial emissions in one county were 100 tons and 10 tons in the other. We appreciate your suggestion to add additional commentary in the introduction to explain why this is important and have now done so.

“We focused on relative emissions changes because, to support equity, emission decreases should be relative to existing emissions levels in each county²¹. That is, places with higher pollution levels should have higher emissions reductions—equal absolute emissions reductions are not equitable if the initial pollution levels are very different in magnitude. For each air pollution sector, our analysis provides information on whether counties’ racial/ethnic and socioeconomic makeup is associated with the magnitude and direction of relative emission changes in the 40 years following the CAA enactment.”

2. One major thing that stood out for me is -- I wonder if the association between racial/SES variables and relative emission changes (the metric being estimated in the paper) needs to be considered together with the associations between racial/SES variables and the absolute emissions. For example, if one sector’s emission was concentrated in one racial group, then what should an “equitable emission reduction strategy” look like? Should we target equal relative reduction (e.g., a reduction by 20% across all counties), OR should we target higher-than-proportional reduction from the high-emitting counties, OR should we target equal absolute reductions (e.g., a reduction of 100 tons) which would then result in a lower-than-proportional reduction for those highly-burden communities? Because of the conceptual challenges above, I find it difficult to understand the associations estimated in the paper. I would suggest 1) a more expanded discussion on the above conceptual challenges in the introduction and the discussion section, and 2) for each panel in figures 2 and 3, show the results of the association between the SES variables and the absolute emissions (perhaps as SI figures).

Thank you for your comment. As we write in our answer to point 1), we support that counties with higher emissions levels should have higher reductions, which is the reasoning behind focusing on the relative emissions change rather than the

absolute change. A 20% decrease in emissions would mean a reduction of 20 tons for a county with initial emissions of 100 tons and a drop of 2 tons for a county with initial emissions of 10 tons. If we focused on absolute changes, we would consider this reduction inequitable for the county with the 2-ton reduction. Although imperfect, using the relative change as an outcome minimizes results that indicate inequities due to counties with high emissions having high absolute emissions decreases. As you point out, an absolute reduction of 100 tons across all counties would ignore initial counties' emissions levels. Additionally, for example, a decrease of 100 tons may not be applicable for minimally polluted areas if their emissions are below 100 tons. We added information to the introduction about the reason for focusing on relative changes (see our answer to point 1).

Regarding adding associations with absolute changes to Figures 2 and 3, these figures are pretty dense already, and adding more information to them would make them even more difficult to read. Our main analysis alone includes 48 models, repeating all of these models with absolute changes presents significant computational limitations. This is an important outcome to consider in future studies.

3. Partly related to the comment above, I find it challenging to fully understand Figures 2 and 3. The shape of the association curve is clear— whether it's linear or not, and the overall directional relationship. But I don't know how to interpret the magnitude of each point on the curve. The figure is hard to interpret partly because of the narrow support of the percentage of each demo group (as shown in the top panel). Would it make more sense to just plot the curve over the range that has the most data coverage for each racial/SES group?

Thank you for your suggestion. We squared (squared root for percent White because the distribution is concentrated in the high percentages values) the x-axis so that areas of the association curves where the data is concentrated can be better visualized. Please note that this was only done for plotting purposes, and the data was not modified.

4. Another important question about this figure is the interpretation of the y-axis. The caption suggests that the y-axis just shows the relative emission changes. If that's the case, are the magnitudes comparable across each row

(for the same sector)? Should the range of the y-axis be identical in the same row? For example, in the second row of figure 2 (industry SO₂ emissions), it seems like the first panel suggests that SO₂ emissions decreased in white-majority counties, but then increased in zero-black counties. I would encourage the authors to examine each panel's normalization baselines (i.e. the y=0 point) and make them comparable and easy to interpret when possible.

Thank you for your comment. The y-axis is the percentage point change in the relative emissions change per increase in the x-axis. The association curves are plotted in reference to the independent variable mean (please see our answers to Reviewer 2 comments 4 and 5). The y-axis is comparable across all sectors because it represents a percentage point. We allow the y-axis range to change depending on the values of the curve so that each curve can be better visualized. If we fixed the y-axis range, some association curves would be small and difficult to see (similar to the issue you pointed out in comment 3). After trying various plotting options, we found this one the easiest to see. The downside is that the reader needs to pay attention to differences in the y-axis range because they vary across plots. Still, within the same sector, the ranges are similar.

Minor:

1. The abstract can be revised to clarify the method and main results. While the expression of “association” is commonly seen in the public health literature, I think a more direct and plain-language description may be more appropriate in the abstract. For example, it could be something like “counties with higher Hispanic population percentages experience a smaller decline in SO₂ emissions from the energy sector”.
2. It will be helpful to briefly explain the method and estimated metrics in the abstract after the first sentence. For example, it could be something like “Here, we calculate the relative emission changes for each county and examine the differential changes across counties with xxx”

Thank you so much for your recommendations! As per your recommendations in points 1-2, we have edited the abstract to make it more accessible to a broader audience.

“We evaluated county-level racial and socioeconomic disparities in emissions changes from six air pollution sectors (industry [SO₂], energy [SO₂, NO_x], agriculture [NH₃], commercial [NO_x], residential [OC], and on-road transportation [NO_x]) in the contiguous United States during the 40 years following the Clean Air Act (CAA) enactment (1970-2010). We calculated relative emission changes and examined the differential changes given county demographics using hi-erarchical nested models. We found racial/ethnic disparities, particularly in the industry and energy sector. We also found that median family income is a significant driver of variation in relative emissions changes in all sectors—counties with high (>\$75K) median family income generally experienced larger relative declines in industry, energy, transportation, residential, and commercial-related emissions. Emissions from most air pollution sectors have, on a national level, significantly decreased following the United States CAA. In this work, we show that the relative reductions in emissions have varied across racial/ethnic and socioeconomic groups.”

3. As the emissions inventory is essential for this paper’s method, I suggest including more descriptions of the CED emissions data. Despite being a popular data source, it might not be clear to everyone what it is. Is this mostly a compilation of NEI and local emissions inventory for the US? This should be in both the methods section and the last paragraph of the introduction section.

Thank you for the suggestions. We have added more details about CED_{GBD-MAP} in the Methods section and Introduction.

Introduction:

“We used county-level data on race/ethnicity and socioeconomic status (SES) from the decennial Census (1970-2010). We leveraged air pollution emissions data from the Community Emissions Data Global Burden of Disease Map (CED_{GBD-MAP}) to estimate county-level emission fluxes for six air pollution sectors, using specific emissions tracers per sector. CED_{GBD-MAP} is an air pollution emissions inventory

that uses emissions data from local and regional inventories and activity data to calculate gridded emissions estimates for the globe from 1970 to 2010. Using hierarchical models, we evaluated disparities in emissions changes by modeling the association between county-level demographics and the decennial relative change in emission fluxes. For each air pollution sector, our analysis provides information on whether counties’ racial/ethnic and socioeconomic makeup is associated with the magnitude and direction of relative emission changes in the 40 years following the CAA enactment.”

Methods section:

“We obtained the emissions data from the Community Emissions Data Global Burden of Disease Map (CED_{GBD-MAP}) emissions inventory³⁸. This is a gridded (0.5° × 0.5° or ~ 55 × 55km) global bottom-up emission inventory from 1970 to 2017. In summary, CED_{GBD-MAP} uses data from various emissions inventories (GAINS, SPEW, US NEI, EDGAR, etc.) and activity data (energy consumption, population, etc.) to calculate global emissions estimates for each chemical compound. For years without available emissions, default estimates are calculated from a linear interpolation and available activity data. Then, local and regional inventories are used to scale sectoral emissions to the national level³⁸.”

4. It is unclear why NO_x is selected as the precursor of the commercial sector. A brief explanation in the methods section (including what are the main emission activities in the commercial sector) would be helpful.

Emissions from the commercial sector specifically refer to commercial combustion, primarily from oil and gas; thus, we are focusing on NO_x emissions. To improve clarity, we added these details to Table 1 (pasted below), which provides some samples of specific sources for each sector. We also clarified that the residential sector refers to residential energy combustion.

Emission Sector	Air pollutant	Primary contributing fuel(s)	Sample activities

Industry	SO ₂	Coal and oil	Industrial boilers in the production of iron-steel, cement, metals, etc.
Energy generation	SO ₂	Coal	Electricity production, fuel production and transportation, oil and gas fugitive/flaring, etc.
	NO _x	Coal, oil, and gas	
Agriculture	NH ₃	N/A	Manure management, soil emissions, enteric fermentation, etc.
Transportation	NO _x	Oil and gas	On-road transportation
Residential energy combustion	OC	Solid biofuel	Cooking, space heating, residential waste burning

Commercial combustion	NO _x	Oil and gas	Combustion in service-providing facilities (e.g., religious facilities, local, state, and federal government, institutional living quarters, sewage treatment facilities, restaurants, and more)
-----------------	-------------	--

5. The results section starts with a discussion of the changes in the demographic structure. This makes me wonder about the role of demographic changes in the association. It would be interesting to see simple sensitivity analyses that calculate the association with fixed demographic patterns. This would then be helpful to understand how demographic changes influence the EJ pattern.

Apologies, we do not fully understand your comment. In this analysis, we evaluated how the demographics of a county in a given decennial (fixed) were associated with relative air pollution emissions changes in the following decennial. This analysis did not evaluate how demographic changes from one decennial to the next affect relative emissions changes. Because we did a nationwide analysis across multiple years and demographic values vary across counties and time, we were able to estimate relative emissions changes across various demographic values (i.e., per one unit change in a specific demographic, what would be the change in the relative emissions). There are many insightful ways to evaluate EJ, and how changes in demographics from one decennial to the following affect air pollution emissions (not changes) would be interesting. In your comment, I believe you are referring to analyzing the association between demographics each year and emissions that same year, which is also a relevant and interesting question. In this study, we intentionally focused on emissions **changes** rather than simple emissions because

we are interested in providing insight into whether *emissions reductions* have been equitable. Modeling emissions and demographics answers an interesting but different research question: Are there inequities in emissions across demographics? Other studies have made this the focus of their work, and results suggest the presence of inequities¹¹. In this study, we specifically decided to focus on emissions changes as that has not been previously evaluated.

¹¹ <https://www.science.org/doi/10.1126/sciadv.abf4491>

REVIEWERS' COMMENTS

Reviewer #1 (Remarks to the Author):

The authors have addressed my comments but I would ask that they add a line in the discussion mentioning that it is likely that spatial variations in emissions do not perfectly align with spatial variations in population exposures (for reasons mentioned in my previous review).

Reviewer #2 (Remarks to the Author):

The authors have done a good job addressing reviewer comments. The manuscript is improved and I recommend it be published. I have a couple of minor comments.

1. Define organic carbon. In the atmospheric science community OC is typically used to refer to the organic portion of particulate matter. However, I suspect here it refers to organic gas or VOC emissions. This needs to be clarified.

"to estimate county-level relative emissions changes for the six air pollution sectors," I am not sure what an air pollution sector is. Presumably it is a source sector?

"However, neither modeled nor measured pollution concentrations provide information about the air pollution sources contributing to the observed disparities" This statement is not correct. There are many source oriented models that track information on emission sources when they calculate concentrations. The work of Tessum is an example of that in the EJ domain.

Reviewer #3 (Remarks to the Author):

I appreciate the authors' efforts in addressing my comments. The current version has significantly improved compared to the prior version and the paper will make an important contribution to the field. I do not have further comments.

Reviewer #1 (Remarks to the Author):

The authors have addressed my comments but I would ask that they add a line in the discussion mentioning that it is likely that spatial variations in emissions do not perfectly align with spatial variations in population exposures (for reasons mentioned in my previous review).

Thank you for your comment, and we apologize for not properly addressing your concern in the first round of revisions. We have now added a sentence about this in the discussion:

“Importantly, emissions are different from air pollution concentrations in geospatial variation. The density of pollution sources spatially might not have as much variation in a hyperlocal spatial resolution (e.g., census tracts or blocks) compared to county-level. This also means that spatial variations in air pollution emissions do not perfectly capture variations in population exposures.”

Reviewer #2 (Remarks to the Author):

The authors have done a good job addressing reviewer comments. The manuscript is improved and I recommend it be published. I have a couple of minor comments.

1. Define organic carbon. In the atmospheric science community OC is typically used to refer to the organic portion of particulate matter. However, I suspect here it refers to organic gas or VOC emissions. This needs to be clarified.

In the CED_{GBD-MAP} emissions inventory, OC refers to particulate organic carbon. We have added this information to the manuscript.

Abstract:

“We evaluate county-level racial/ethnic and socioeconomic disparities in emissions changes from six air pollution source sectors (industry [SO₂], energy [SO₂, NO_x], agriculture [NH₃], commercial [NO_x], residential [particulate organic carbon], and on-road transportation [NO_x]) in the contiguous United States during the 40 years following the Clean Air Act (CAA) enactment (1970-2010).”

Introduction:

“In this work, we evaluated county-level racial/ethnic and socioeconomic disparities in air pollution emissions changes in the contiguous US from 1970 to 2010 in the transportation (nitrogen oxides [NO_x]), agriculture (ammonia [NH₃]), residential (particulate organic carbon), commercial (NO_x), industry (sulfide dioxide [SO₂]), and energy (NO_x and SO₂) sectors.”

Results:

“On average, air pollution emissions across the US decreased substantially from 1970 to 2010 from all source sectors we considered except for agriculture NH₃ and residential particulate organic carbon (OC)”

"to estimate county-level relative emissions changes for the six air pollution sectors," I am not sure what an air pollution sector is. Presumably it is a source sector?

Thank you for your comment. Yes, you are correct in that by air pollution sector, we mean air pollution source sector. We have edited the language throughout the manuscript to clarify this point.

"However, neither modeled nor measured pollution concentrations provide information about the air pollution sources contributing to the observed disparities" This statement is not correct. There are many source oriented models that track information on emission sources when they calculate concentrations. The work of Tessum is an example of that in the EJ domain.

Sorry for the confusion. What we meant is that analyzing inequities in air pollution concentrations alone does not provide information about air pollution sources. We agreed that air pollution emissions could be used to model PM_{2.5} concentrations and, subsequently, inequities, as in Tessum et al.

We have edited the text so we communicate this point more clearly:

“However, pollution concentrations alone do not provide information about specific air pollution sources contributing to the observed disparities. A couple of studies have addressed this knowledge gap by modeling source-specific air pollution concentrations.^{7,13.}”

Reviewer #3 (Remarks to the Author):

I appreciate the authors' efforts in addressing my comments. The current version has significantly improved compared to the prior version and the paper will make an important contribution to the field. I do not have further comments.

Thank you!